# Briarenones A‒C, New Briarellin Diterpenoids from the Gorgonian *Briareum violaceum*

**DOI:** 10.3390/md17020120

**Published:** 2019-02-17

**Authors:** Yang Cheng, Atallah F. Ahmed, Raha S. Orfali, Chang-Feng Dai, Jyh-Horng Sheu

**Affiliations:** 1Department of Marine Biotechnology and Resources, National Sun Yat-sen University, Kaohsiung 804, Taiwan; jack1991106@yahoo.com.tw; 2Department of Pharmacognosy, College of Pharmacy, King Saud University, Riyadh 11451, Saudi Arabia; afahmed@ksu.edu.sa (A.F.A.); rorfali@ksu.edu.sa (R.S.O.); 3Department of Pharmacognosy, Faculty of Pharmacy, Mansoura University, Mansoura 35516, Egypt; 4Institute of Oceanography, National Taiwan University, Taipei 112, Taiwan; corallab@ntu.edu.tw; 5Graduate Institute of Natural Products, Kaohsiung Medical University, Kaohsiung 807, Taiwan; 6Frontier Center for Ocean Science and Technology, National Sun Yat-sen University, Kaohsiung 804, Taiwan; 7Department of Medical Research, China Medical University Hospital, China Medical University, Taichung 404, Taiwan

**Keywords:** *Briareum violaceum*, briarenones, briarellin, gorgonian

## Abstract

Three new eunicellin-derived diterpenoids of briarellin type, briarenones A‒C (**1**‒**3**), were isolated from a Formosan gorgonian *Briareum violaceum*. The chemical structures of the compounds were elucidated on the basis of extensive spectroscopic analyses, including two-dimensional (2D) NMR. The absolute configuration of **1** was further confirmed by a single crystal X-ray diffraction analysis. The in vitro cytotoxic and anti-inflammatory potentialities of the isolated metabolites were tested against the growth of a limited panel of cancer cell lines and against the production of superoxide anions and elastase release in *N*-formyl-methionyl-leucyl-phenyl-alanine and cytochalasin B (fMLF/CB)-stimulated human neutrophils, respectively.

## 1. Introduction

Gorgonians belonging to genus *Briareum* (phylum Cnidaria, family Briareidae) are considered to be a rich source of highly oxygenated diterpenoids, particularly those derived from briarein (briarane) [1,2,3,4,5,6,7,8,9,10,11], eunicellin (cladiellin) [1,12,13,14,15,16], asbestinin [11,15,17,18,19], and, to a lesser extent, cembrane [20]. These metabolites were shown to exhibit various bioactivities such as anti-inflammatory [2,4,5,13,21], analgesic [22], cytotoxic [7,8,9,10,14,15,23], antimalarial [12,15], antiviral [10,15], and antimicrobial [15] activities. Briarellins constitute a class of tetracyclic oxygenated diterpenoids in which an additional seven-membered ether ring (oxepane) is formed between C-3 and C-16 of the eunicellin skeleton. These metabolites were originally discovered from Caribbean gorgonians of genus *Briareum*, e.g., *B. asbestinum* [13,14,16] and *B. polyanthes* [12,15], and also showed interesting bioactivities. Yet, briarellins remain to be isolated from the Pacific *Briareum* genus, although briarellins with the α,β-conjugated enone in the six-membered ring analogs were previously reported from one species of a taxonomically linked *Pachyclavularia* genus [24,25,26]. This prompted us to extensively investigate the chemical constituents of a gorgonian *Briareum violaceum* growing wildly in the Taiwanese waters and to screen their cytotoxic and anti-inflammatory activities. The current study led to the discovery of a group of three new briarellin diterpenoids, briarenones A‒C (**1**‒**3**), characterized by having *cis* fused cyclohexane, cyclodecane, and oxepane rings. The chemical structures of these molecules were resolved by various spectroscopic analyses, including two-dimensional (2D) NMR correlation analysis, while their absolute configurations were assigned by single-crystal X-ray diffraction analysis for **1**.

## 2. Results and Discussion

The lyophilized specimen of the gorgonian was extracted with ethyl acetate (EtOAc). The chromatographic fractionation of the solvent-free extract on silica gel and reversed-phase C18 columns and the final separation using reversed-phase high-performance liquid chromatography (RP-HPLC) afforded diterpenoids **1**‒**3** (Figure 1; Appendix A).

Briarenone A (**1**) was obtained as a needle crystal, ${\left[\alpha \right]}_{D}^{25}$ +224.4 (CHCl_3_). Its molecular formula C_22_H_32_O_6_ was determined by the sodium adduct peak [M + Na]^+^ at *m/z* 415.2089 in the high-resolution electrospray ionization mass spectrometry (HRESIMS), inferring seven degrees of unsaturation. The infrared (IR) absorption bands at *ν*_max_ 3477, 1728, and 1670 cm^−1^ indicated the presence of hydroxyl, ester carbonyl, and α,β-unsaturated carbonyl functionalities, respectively. As the NMR spectra of **1** showed numerous broad and very weak signals due to the likely presence of a mixture of slowly interconverting conformers on the NMR time scale, remeasuring the spectra at −10 °C allowed us to have better resolution for the signals. The ^1^H NMR spectrum displayed the signals of four methyls, including one olefinic (δ_H_ 1.99 ppm, s), two methyls attached to oxygen-bearing carbons (δ_H_ 1.27 ppm, s and 1.28 ppm, s), and one secondary methyl (δ_H_ 1.09 ppm, d, *J* = 7.2 Hz), signifying the terpenoid nature of **1**. Furthermore, the NMR data (Table 1) indicated the existence of an acetyl (δ_C_ 171.3 ppm, δ_C_/δ_H_ 21.5/2.08 ppm) and an α,β-unsaturated ketone (δ_C_ 198.0, 157.0, and δ_C_/δ_H_ 128.0/5.95 ppm). Taking into account the unsaturation degrees mentioned above, the 22 carbon signals in the ^13^C NMR spectrum of **1** (Table 1) are, thus, ascribable to the presence of a tetracyclic diterpenoid acetate. Three ring-juncture methines (δ_C_/δ_H_ 51.1/2.71, 48.3/2.5, and 36.8/3.13 ppm), two tetrahydrofuran (THF)-oxymethines (δ_C_/δ_H_ 83.4/3.72 and 77.9/4.79 ppm), and one oxymethylene (δ_C_/δ_H_ 63.7/3.62 and 3.34 ppm) suggested a briarellin-related [13,16,24,25] or an asbestinin-related [15,17,18] structure for **1**. However, since the protons of two methyl groups (δ_H_ 1.99 ppm, s, H_3_-20 and 1.09 ppm, d, *J* = 7.2 Hz, H_3_-17), but not one methyl group, showed *^3^J_CH_* correlations in the heteronuclear multiple bond correlation (HMBC) spectrum with two of the angular methine carbons (δ_C_ 51.1 and 48.3 ppm for C-10 and C-14, respectively), the briarellin-type structure was designated for **1** (Figure 2). Three partial structures were assigned by the analysis of proton correlation spectroscopy (^1^H-^1^H COSY), including that assigned by the allylic correlation of the H_3_-20 (δ_H_ 1.99 ppm) with the olefinic proton H-12 (δ_H_ 5.95 ppm, s) (Figure 2). The connectivity of these partial structures, the positions of the hydroxyl, acetoxyl, and ketone carbonyl, and the ring-fusion carbons of THF and oxepane rings were deduced from the complete correlation analyses of the HMBC spectrum as illustrated in Figure 2. Furthermore, a comparison of the ^13^C NMR data of **1** with those of pachyclavulariaenone F (**4**), isolated from *Pachyclavularia violacea* [25], verified the replacement of the C-4 hydroxymethine group (δ_C_ 69.7 ppm, CH) in **4** by a methylene group (δ_C_ 33.6 ppm, CH_2_) in **1**. This was associated with a significant upfield chemical shift at C-5 (∆δ_C_ ‒8.5 ppm) in **1** relative to that in **4**. The planar structure of **1** was accordingly established (Figure 2). The relative configuration of **1** at C-1, C-2, C-3, C-6, C-7, C-9, C-10, C-14, and C-15, was assigned by the analysis of nuclear rotating-frame Overhauser effect spectroscopy (ROESY) correlations (Figure 2), which were found to be consistent with that of **4** as illustrated in Figure 2. Consequently, two opposite sets of nuclear Overhauser effect (NOE) interactions were observed for **1**. One set displayed NOEs for H-1/H-10, H-1/H-14, H-1/H_3_-17, H-14/H_3_-17, H-1/H-8β (δ_H_ 2.03 ppm, d, *J =* 15.0, 12.0 Hz), and H-8β/H_3_-19, and another set exhibited NOEs for H-2/H_3_-18 and H_3_-18/H-6, while H-9 did not show any NOE correlation with H-10. Thus, H-1, H-10, H-14, H_3_-17, and H_3_-19 should be positioned in the β-face of the molecule, whereas H-2, H-9, H_3_-18, and H-6 should be positioned in its α-face. In order to confirm the molecular structure of **1**, including the absolute configuration, a single-crystal X-ray structure analysis was further performed (Figure 3). The compound was slowly crystallized from MeOH as a monohydrate C_22_H_32_O_6_·H_2_O, with H-bond holding the H_2_O molecule in a hydrophilic pocket. From a dataset collected using copper radiation, the X-ray crystallographic analysis (Appendix A) of **1** determined the absolute configurations of briarenone A (**1**) as 1*S*,2*R*,3*R*,6*S*,7*S*,9*R*,10*R*,14*S*,15*S*, which forms a hydrogen bond from 7-OH with a molecule of water.

Briarenone B (**2**) was obtained as a white powder. It possessed a molecular formula of C_20_H_28_O_4_, as established from the sodiated ion peak in the HREIMS (*m/z* 355.1880 [M + Na]^+^) of **2**, accounting for seven degrees of unsaturations. The IR absorptions at *ν*_max_ 3420 and 1670 cm^−1^ indicated the presence of hydroxyl and conjugated carbonyl functionalities, respectively. The NMR data of **2**, measured in CDCl_3_ at −10 °C, were comparable to those of the framework of **1** except in the replacement of an *sp^3^* oxycarbon (δ_C_ 74.6 ppm, C, C-7) and a methyl group (δ_C_/δ_H_ 23.7/1.28 ppm, CH_3_, CH_3_-19) in **1** by an exomethylene group (δ_C_/δ_H_ 148.3 ppm, C and 117.7/5.35 and 5.18 ppm, CH_2_, respectively) in **2**. Analysis of NMR data of **2** (Table 1) and correlations found in the ^1^H-^1^H COSY and HMBC spectra (Figure 4) enabled the establishment of the gross structure of **2**. Furthermore, the observed NOE correlations for **2** (Figure 5) assigned similar β-positioning for H-1, H-10, H-14, and H_3_-17, and α-orientation for H-2, H-9, and H_3_-18 as found in **1**. Moreover, one of the exomethylene protons (δ_H_ 5.18 ppm, s) exhibited NOE interaction with H-10 (δ_H_ 2.97 ppm, br d, *J* = 7.0 Hz), while the other one (δ_H_ 5.35, s) displayed NOE with the hydroxymethine proton at C-6 (δ_H_ 4.07 ppm, dd, *J* = 11.0, 4.5 Hz). A molecular modeling investigation revealed that H-6 did not exhibit NOE interaction in the nuclear Overhauser effect spectroscopy (NOESY) with the methylene protons at C-8 (distances >3.7 Å). This designated the α-orientation of 6-OH and, hence, the 6*R* configuration as shown in Figure 6. Moreover, it was found that eunicellins with similar substitutions in the ten-membered ring, but with an 6β-OH (Figure 6), displayed distinctive chemical shifts at C-6 (δ_C_ < 74.0 ppm), C-7 (δ_C_ ≥ 150.0 ppm), and H-6 (δ_H_ > 4.25 ppm) [27,28,29,30] than those assigned for **2** (δ 78.3, 148.3, and 4.07 ppm, respectively). This observation also suggests the α-orientation of 6-OH in **2**. On the basis of the above findings, the absolute configuration of **1**, and the biogenetic consideration, the configuration 1*S*, 2*R*,3*R*,6*R*,9*R*,10*R*,14*S*,15*S* was, thus, established for **2**.

The HREIMS (*m*/*z* 355.1880 [M + Na]^+^) and NMR data (Table 1) indicated a molecular formula C_20_H_28_O_4_ for briarenone C (**3**), inferring that **3** is an isomer of **2**. Therefore, it was found that **3** possessed an α,β-unsaturated carbonyl (IR: *ν*_max_ 1668 cm^−1^ δ_C_ 198.1 ppm, C, 156.4 ppm, C, and δ_C_/δ_H_ 128.6/5.91 ppm, CH) and a hydroxy methine group (IR: *ν*_max_ 3299 cm^−1^; δ_C_/δ_H_ 65.0/4.84 ppm, CH). However, compound **3** differs in the presence of a trisubstituted double bond (δ_C_/δ_H_ 135.4/5.52 ppm, CH and δ_C_ 128.8 ppm, C), instead of a 1,1-disubstituted double bond in **2**, with the appearance of an olefinic methyl (δ_C_/δ_H_ 29.2/1.93 ppm). The hydroxyl group was found to be linked to the allylic carbon (C-5) due to the HMBC correlation observed from the olefinic methyl (δ_H_ 1.93 ppm, H_3_-19) to the *sp^2^* methine carbon (δ_C_ 135.4 ppm, C-6), which in turn exhibited ^1^H-^1^H COSY correlation with H-5 (δ_H_ 4.84 ppm, dd, *J* = 8.5, 8.5 Hz). Moreover, comparison of NMR data of **3** with those of pachyclavulariaenone D pointed out that **3** is the 5-hydroxy isomer of pachyclavulariaenone D [25]. The gross structure of **3** was further elucidated from the full 2D NMR spectroscopic correlation analyses of **3** (Figure 4). The NOE interaction of H_3_-19 with the olefinic methine protons H-6 (δ_H_ 5.52 ppm, d, *J* = 8.5 Hz) (Figure 5), in addition to the downfield shift of C-19 (δ_C_ 29.2 ppm), indicated the *Z* geometry of the 6,7-double bond [31]. Moreover, the NOE correlations for H-2/H_3_-18, H_3_-18/H-5, H-5/H-8α (δ_H_ 2.74 ppm, d, *J* = 14.5 Hz), and H-8α/H-9 disclosed the 5*S* configuration. These NOEs were further validated by a molecular modeling study (Figure 6). Finally, full NOE correlation analysis (Figure 5) coupled with the previous identification of the absolute configuration of **1**, which was co-isolated along with **3**, designated the 1*S*,2*R*,3*R*,5*S*,9*R*,10*R*,14*S*,15*S* configuration for **3**.

Although briarellin-type diterpenoids (e.g., briarellins A–S) were discovered initially from Caribbean gorgonians of genus *Briareum* [12,13,14,15,16], since 1995, numerous related analogs were successfully isolated and identified from *Pachyclavularia violacea* inhabiting Taiwanese [24,25] and Indonesian [26] waters. Unlike the previously identified briarellins, the structures of briarenones A–C (**1**–**3**), isolated in this study from *B. violaceum*, were found to be similar to those of pachyclavulariaenones [24,25] in possessing the three ring-juncture protons (H-1, H-10, and H-14) *cis* to each other and forming the *cis* fusion of the cyclohexane, cyclodecane, and oxepane rings. Also, the determination of the absolute configurations of **1**–**3** by NOE correlation analysis coupled with X-ray crystallographic analysis of **1** using copper radiation, could imply the absolute configurations for similar briarellins isolated from the genera of *Briareum* and *Pachyclavularia*, such as pachyclavulariaenones A–G [24,25], to be similar to that of **1**. Compounds of this kind possessing a six-membered carbocyclic ring *cis*-fused to both of the ten-membered carbocyclic and seven-membered ether rings, making three ring-junction protons (H-1, H-10, and H-14) *cis* to each other, including those reported from *Pachyclavularia violacea* [24,25,26], can be considered as a useful chemotaxonomic marker in classification of gorgonians and in approaching a resolution of the argument for considering the gorgonian species of *Pachyclavularia* to be the same as those of genus *Briareum* [26,32].

The isolated compounds were evaluated against the growth of DLD-1 (human colon adenocarcinoma), HT-29 (human colon carcinoma), and HuCC-T1 (human colon cholangiocellular carcinoma) cell lines. The in vitro anti-inflammatory activities of compounds **1**‒**3** were also measured against the release of elastase and the production of superoxide anions in *N*-formyl-methionyl-leucyl-phenyl-alanine and cytochalasin B (fMLF/CB)-activated neutrophils. However, the compounds did not show either cytotoxic or anti-inflammatory activities in the tested in vitro models.

## 3. Materials and Methods

### 3.1. General Experimental Procedures

Optical rotations and IR spectra were measured on a JASCO P-1020 polarimeter and a JASCO FT/IR-4100 (JASCO Corporation, Tokyo, Japan) spectrophotometer, respectively. HRESIMS spectra were measured on a Bruker APEX II mass spectrometer (Bruker, Bremen, Germany). ^1^H and ^13^C NMR spectra were obtained on a Varian Unity INOVA 600 FT-NMR (or Varian Unity INOVA500 FT-NMR) instruments (Varian Inc., Palo Alto, CA, USA) at 600 MHz (or 500 MHz) for ^1^H, and 150 MHz (or 125 MHz) for ^13^C in CDCl_3_. Thin-layer chromatography (TLC) analyses were performed on precoated silica (Si) gel plates (Kieselgel 60 F-254, 0.2 mm), and Si gel (230–400 mesh) (Merck, Darmstadt, Germany) and C18-reversed phase Si gel (RP-18; 40–63 µM) (Parc-Technologique BLVD, Quebec, Canada) were used for column chromatography. Further purification and the isolation of compounds were performed by reversed-phase high-performance liquid chromatography (RP-HPLC) on a Hitachi L-2455 HPLC apparatus with a Supelco C18 column (250 × 21.2 mm, 5 μm).

### 3.2. Animal Material

The soft coral *B. violaceum* was collected from Jihui Fish Port (Taitung, Taiwan) by scuba divers at a depth of 10–15 m on March 2013 and stored in a freezer at −20 °C until extraction. Moreover, the soft coral was taxonomically identified by Prof. Chang-Feng Dai, National Taiwan University, Taipei. A voucher specimen was deposited in the Department of Marine Biotechnology and Resources, National Sun Yat-sen University, Kaohsiung.

### 3.3. Extraction and Isolation

The freeze-dried soft coral (0.5 kg) was extracted with EtOAc (3 × 3 L) and filtered. The filtrate was evaporated in vacuo to yield the EtOAc extract (3.90 g). The extract was fractionated by Si gel column chromatography using EtOAc in *n*-hexane (6.25% to 100%) as a gradient elution system to afford 21 fractions (A01–A21). Fraction A04 was further fractionated on an RP-18 Si gel column, using MeOH-H_2_O (2.5:1) as a mobile phase, into eight subfractions (A0401–A0408). Compound **1** (7.4 mg) was obtained from A405 after two-step purification on RP-HPLC using MeOH-H_2_O (3:2) and acetyl nitrite (CH_3_CN)-H_2_O (1:2). Subfractions A402 and A403 were combined together on the basis of their similar TLC chromatogram and were further separated on RP-HPLC using CH_3_CN-H_2_O (1:2) to yield compounds **2** (4.5 mg) and **3** (2.2 mg).

#### 3.3.1. Briarenone A (**1**)

Colorless needles; ${\left[\alpha \right]}_{D}^{25}$ +224.4 (*c* 7.4, CHCl_3_); IR (neat) *ν*_max_ 3477, 2964, 2929, 1728, 1670, and 756 cm^−1^; ^1^H and ^13^C NMR data (600/150 MHz; CDCl_3_), Table 1; ESIMS *m/z* 415 [M + Na]*^+^*; HRESIMS *m/z* 415.2089 [M + Na]^+^ (calculated for C_22_H_32_O_6_Na, *m/z* 415.2091).

#### 3.3.2. Briarenone B (**2**)

Colorless powder; ${\left[\alpha \right]}_{D}^{25}$ −10.7 (*c* 4.5, CHCl_3_); IR (neat) *ν*_max_ 3420, 2912, 2851, 1670, and 772 cm^−1^; ^1^H and ^13^C NMR data (500/125 MHz; CDCl_3_), Table 1; ESIMS *m/z* 355 [M + Na]^+^; HRESIMS *m/z* 355.1880 [M + Na]^+^ (calculated for C_20_H_28_O_4_Na, *m/z* 355.1880).

#### 3.3.3. Briarenone C (**3**)

Colorless powder; ${\left[\alpha \right]}_{D}^{25}$ −14.8 (*c* 2.2, CHCl_3_); IR (neat) *ν*_max_ 3299, 2921, 2851, and 1668 cm^−1^; ^1^H and ^13^C NMR data (500/125 MHz; CDCl_3_), Table 1; ESIMS *m/z* 355 [M + Na]^+^; HRESIMS *m/z* 355.1880 [M + Na]^+^ (calculated for C_20_H_28_O_4_Na, *m/z* 355.1880).

#### 3.3.4. Single-Crystal X-Ray Crystallography of **1**

A suitable colorless prism of compound **1** was obtained from a solution in MeOH by slow evaporation for a month at 4 °C. The crystal (0.20 × 0.18 × 0.17 mm^3^) was analyzed at 100(2) K, space group *P*2_1_2_1_2_1_ (# 19). Cell: *a* = 8.3456(3) Å, *b* = 10.6813(3) Å, *c* = 23.4466(8) Å, *V* = 2090.07(12) Å^3^, *Z* = 4, *D*_calcd_ = 1.305 mg/m^3^, and *μ*(CuKα) = 0.790 mm^−1^. Intensity data of single-crystal X-ray diffraction were measured on a Bruker APEX DUO diffractometer. Of the 13326 reflections collected, only 3658 independent reflections [*R(int)* = 0.0331] with I > 2σ(I) were used for the analysis. The structure was solved by direct method and refined by a full-matrix least squares on F2 method. The refinement converged to a final *R_1_* = 0.0401, *wR_2_* = 0.1064, with goodness-of-fit = 1.091. For coordinates corresponding to the absolute stereochemistry represented, absolute structure parameter 0.04(6) was obtained [33].

### 3.4. Bioassays

#### 3.4.1. Cytotoxicity Assay

Cancer cell (DLD-1, HT-29, and HuCC-T1) lines were purchased from the American Type Culture Collection (ATCC). Compounds **1**‒**3** were evaluated for cytotoxic activity using an Alamar blue assay. Alamar Blue (resazurin) is an important non-toxic redox indicator that is utilized to assess metabolic function and cell health. A complete description of this test was previously described [34,35]. The absorbance was measured at 570 nm using an ELISA plate reader (Thermo Fisher Scientific Instruments Co., Ltd., Vantaa, Finland).

#### 3.4.2. In Vitro Anti-Inflammatory Assays

Human neutrophils were obtained from blood by dextran sedimentation, Ficoll-Hypaque centrifugation, and hypotonic lysis and then incubated as previously described [36]. Neutrophils (6 × 10^5^ cells∙mL^−1^) incubated at 37 °C in Hank’s balanced salt solution (HBSS) with MeO-Suc-Ala-Ala-Pro-Val-*p*-nitroanilide (100 μM) and Ca^2+^ (1 mM) were treated with dimethyl sulfoxide (DMSO) or the tested compound for 5 min. The activation of neutrophils was challenged for 10 min with fMLF (100 nM)/CB (0.6 and 0.5 μg∙mL^−1^ for superoxide anion generation and elastase release, respectively). The anti-inflammatory activities of compounds **1**‒**3** were measured with examining the inhibition of fLMF/CB-induced human neutrophils producing superoxide anion and elastase, using UV spectrometer detection at wavelengths of 550 nm and 405 nm, respectively [36,37].

## 4. Conclusions

Three new briarellin type diterpenoids, briarenones A‒C (**1**‒**3**), were identified from *Briareum violaceum* inhabiting Taiwanese waters. The compounds have three *cis* ring-juncture protons (H-1, H-10, and H-14) due to the *cis* fusion of the cyclohexane, cyclodecane, and oxepane rings. The molecular structures of this type may be considered as a useful chemotaxonomic marker in the identification of some species of genus *Briareum*. The absolute configurations of the compounds were assigned on the basis NOE correlation analysis coupled with a single-crystal X-ray diffraction analysis for briarenone A. The isolated metabolites showed no in vitro cytotoxicity against DLD-1, HT-29, and HuCC-T1 cells and did not inhibit the superoxide anion generation or elastase release in fMLF/CB-stimulated neutrophils. Compounds **1**‒**3** did not exhibit cytotoxic and anti-inflammatory activities in this study; however, more biological activity screening should be carried out to discover their pharmaceutical potential. Moreover, the absolute configuration of **1** analyzed by X-ray diffraction would be useful for elucidation of the structurally similar metabolites isolated from the genera *Briareum* and *Pachyclavularia*.

## Figures and Tables

**Figure 1 marinedrugs-17-00120-f001:**
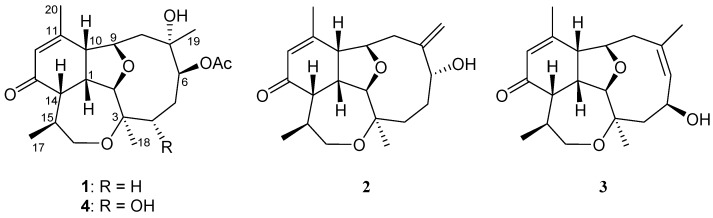
Structures of new briarellin diterpenoids (**1**‒**3**) isolated from *Briareum violaceum* and pachyclavulariaenone F (**4**).

**Figure 2 marinedrugs-17-00120-f002:**
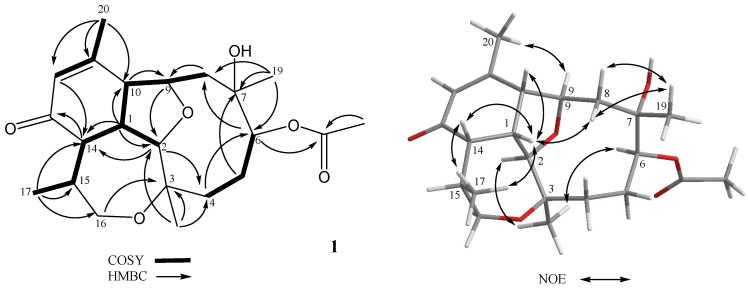
Key ^1^H-^1^H COSY, HMBC, and NOE correlations for **1****.**

**Figure 3 marinedrugs-17-00120-f003:**
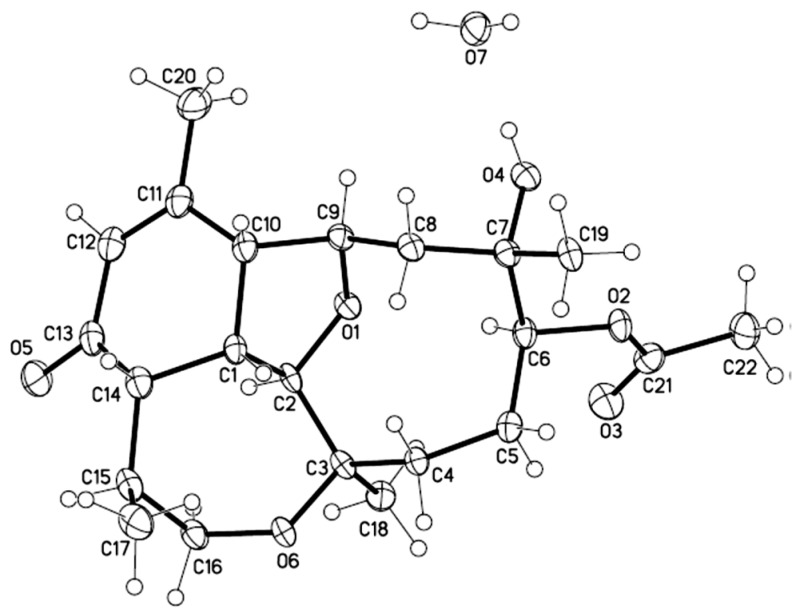
Oak Ridge Thermal Ellipsoid Plot (ORTEP) diagram of the molecular structure of **1** as determined by X-ray analysis.

**Figure 4 marinedrugs-17-00120-f004:**
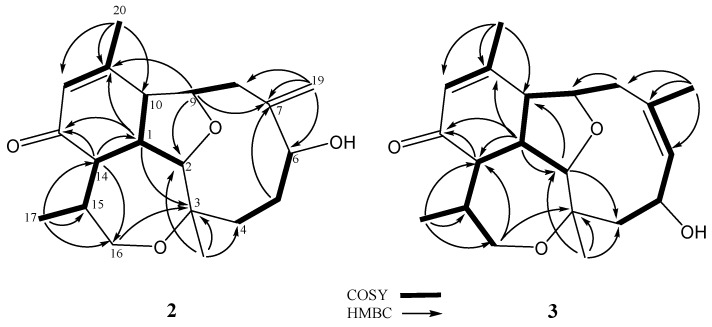
Key ^1^H-^1^H COSY and HMBC correlations for **2** and **3**.

**Figure 5 marinedrugs-17-00120-f005:**
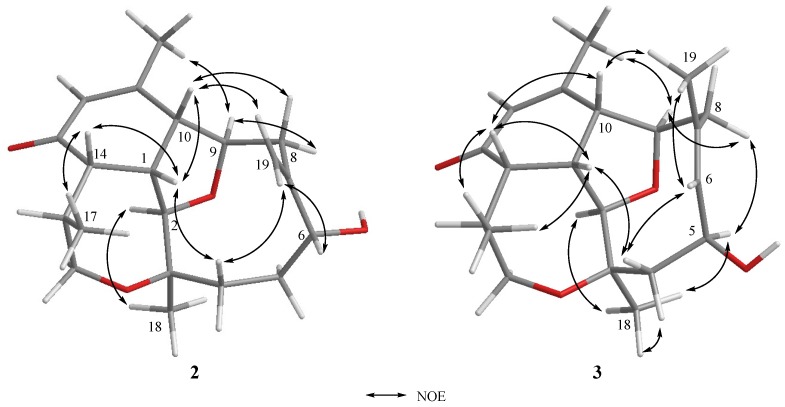
Key NOE correlations for **2** and **3**.

**Figure 6 marinedrugs-17-00120-f006:**
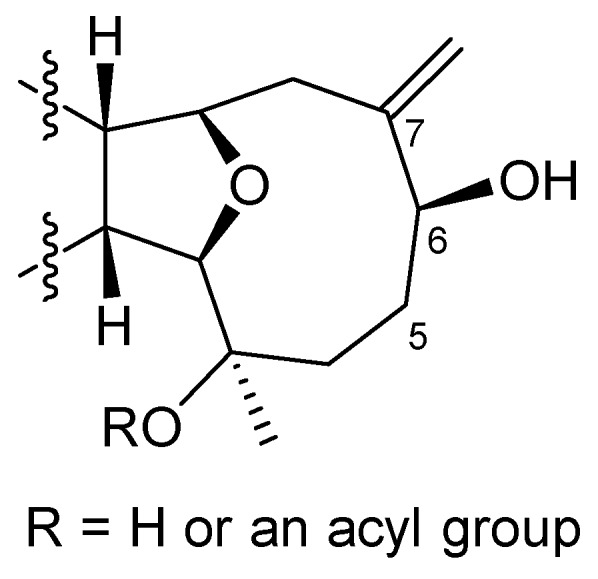
A partial structure of eunicellin-derived diterpenoids: cladiellisin [27], 3-acetyl cladiellisin [28], pachycladin B [29], and klysimplexins C and E [30].

**Table 1 marinedrugs-17-00120-t001:** The ^1^H and ^13^C NMR chemical shifts for **1**‒**3**.

	1		2		3	
#	δ_H_, m (*J* in Hz) *^a^*	δ_C_ *^b^*, type	δ_H_, m (*J* in Hz) *^c^*	δ_C_ *^d^*, type	δ_H_, m (*J* in Hz) *^c^*	*δ*_C_*^d^*, type
1	3.13 ddd (9.6, 5.4, 4.8)	36.8, CH	3.12 ddd (10.0, 7.0, 4.5)	37.7, CH	2.89 m	40.4, CH
2	3.72 d (9.6)	83.4, CH	3.76 d (10.0)	86.4, CH	3.62 d (10.5)	85.8, CH
3		76.5, C		75.4, C		75.5, C
4α	1.94 br d (10.8)	33.6, CH_2_	1.58 m	31.6, CH_2_	1.72 d (15.0)	46.4, CH_2_
4β	1.49 m		1.58 m		2.36 dd (14.5, 8.5)	
5α	1.94 br d (10.8)	27.2, CH_2_	2.04 m	28.7, CH_2_	4.84 dd (8.5, 8.5)	65.0, CH
5β	1.58 ddd (11.4, 10.8, 3.6)		1.79 m			
6	5.77 dd (11.4, 3.6)	78.0, CH	4.07 dd (11.0, 4.5)	78.3, CH	5.52 d (8.5)	135.4, CH
7		74.6, C		148.3, C		128.8, C
8α	1.81 dd (15.0, 3.6)	45.5, CH_2_	2.77 dd (14.0,4.5)	37.0, CH_2_	2.74 d (14.5)	38.6, CH_2_
8β	2.03 dd (15.0, 12.0)		2.14 dd (14.0,5.0)		2.01 dd (14.5, 3.5)	
9	4.79 dd (12.0, 3.6)	77.9, CH	4.45 dd (5.0, 4.0)	82.2, CH	4.29 dd (6.5, 3.0)	81.2, CH
10	2.71 br d (6.0)	51.1, CH	2.97 br d (7.0)	50.3, CH	2.82 br d (8.5)	48.3, CH
11		157.0, C		156.6, C		156.4, C
12	5.95 s	128.0, CH	5.93 s	127.4, CH	5.91 s	128.6, CH
13		198.1, C		198.1, C		198.1, C
14	2.50 br d (4.8)	48.3, CH	2.42 br d (4.5)	48.5, CH	2.31 dd (4.5, 5.5)	48.4, CH
15	2.67 m	30.0, CH	2.59 m	30.4, CH	2.58 m	32.3, CH
16α	3.62 d (13.8)	63.8, CH_2_	3.35 dd (13.5,3.5)	63.9, CH_2_	3.52 m	65.2, CH_2_
16β	3.34 br d (13.8)		3.54 d (13.0)		3.52 m	
17	1.09 d (7.2)	17.2, CH_3_	1.04 d (7.5)	17.2, CH_3_	1.02 d (7.5)	18.5, CH_3_
18	1.27 s	22.2, CH_3_	1.25 s	23.8, CH_3_	1.41 s	27.5, CH_3_
19	1.28 s	23.7, CH_3_	5.35 s; 5.18 s	117.7, CH_2_	1.93 s	29.2, CH_3_
20	1.99 s	21.9, CH_3_	1.98 s	21.8, CH_3_	1.91 s	21.9, CH_3_
Ac	2.08 s	21.5, CH_3_				
		171.3, C				

*^a^* Spectra recorded at 600 MHz in CDCl_3_ at −10 °C; *^b^* spectra recorded at 150 MHz in CDCl_3_ at −10 °C; *^c^* spectra recorded at 500 MHz in CDCl_3_; *^d^* spectra recorded at 125 MHz in CDCl_3_.

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
