# Peer review of "Briarenones A‒C, New Briarellin Diterpenoids from the Gorgonian *Briareum violaceum"

_marinedrugs, 2019, doi:10.3390/md17020120_

Round 1
Reviewer 1 Report
Dear Sirs
Due to low quality of Figures presented in Supplementary I am not able to review this manuscript. The Figures are unreadable when printed and of very low quality (pixel form) when presented on computer screen, even after zoom. There is lack of Figure S10 (13 NMR spectrum for the compound 2). In Figures S2 and S9 (1H NMR for the compounds 1 and 2), the chemical shifts of majority signals are not presented. Therefore, I am not able to compare the information presented in the main text and in Table 1 and in the Figures in Supplementary. Additionally, there is a difference in some decimal places between chemical shift in Tables 1 and in the spectra, which makes review tedious. I suppose that the X-ray data help to assign properly the NMR signals and to determine the structures. However, it must be clearly presented. Of course, I am ready to review this manuscript again, but the quality must be considerably improved.
Author Response
We thank the reviewer for raising the following points and for his valuable comments and kind suggestions.
Comments | Responses |
Due to the low quality of Figures presented in Supplementary, I am not able to review this manuscript. The Figures are unreadable when printed and of very low quality (pixel form) when presented on a computer screen, even after zoom. | We have replaced the low-resolution 1H NMR, 13CNMR spectra by the high-resolution version. DEPT spectra of the three compounds were also added. |
There is a lack of Figure S10 (13C NMR spectrum for the compound 2). In Figures S2 and S9 (1H NMR for the compounds 1 and 2), the chemical shifts of majority signals are not presented. Therefore, I am not able to compare the information presented in the main text and in Table 1 and in the Figures in Supplementary. | The 13C NMR spectrum for the compound 2 (Figure S12) is now included (in detailed sections) along with the DEPT spectra (Figure S13). Moreover, the 1H NMR spectrum of each compound is divided into low and high field regions for better reading of the 1H signal splitting pattern and chemical shifts. |
Additionally, there is a difference in some decimal places between the chemical shift in Tables 1 and in the spectra, which makes review tedious. I suppose that the X-ray data help to assign properly the NMR signals and to determine the structures. However, it must be clearly presented. | Chemical shifts in Tables 1 were revised to be matched with those in the spectra (the decimal places of 13C NMR chemical shifts were also adjusted). Also, we think that the NMR data for compounds of this type are sufficient for us to elucidate the chemical structures. X-ray data were used to confirm the structures elucidated by NMR and other spectroscopic data. |

Reviewer 2 Report
The manuscript by Cheng et al describes the isolation and structure elucidation of three new diterpenoids from the gorgonian Briareum violaceum. A crystal structure of one of the compounds defined its absolute stereochemistry and suggested the stereochemistry of the other two. Assays for cancer cell cytoxicity and anti-inflammatory activity showed no activity.
The work is thorough, well performed and described with only minor English corrections suggested (see below). I would like to see the mixing times for the NOESY experiments included in the supplementary material.
line 21 replace "compounds with "the compounds"
line 38 replace "have been originally discovered" with "were originally isolated"
line 85 replace "HMBC" with "the HMBC"
line 99 replace "has been " with "was"
line 110 and 126 replace "molecule" with "molecular"
line 122 replace "NOE" with "an NOE"
line 127 replace "for " with "of"
line 158 replace "This" with "These"
line 192 replace "frozen-dried" with "freeze-dried"
line 192 replace ", filtered" with "and filtered"
line 193 replace "over" with "by"
line 219 replace "for the analysis" with "were used for the analysis"
line 242 remove italic formatting on "inhabiting"
line 243 replace "This" with "These"
Author Response
We thank the reviewer for raising the following points and for his valuable comments and kind suggestions.
Comments | Responses |
The work is thorough, well performed and described with only minor English corrections suggested (see below). I would like to see the mixing times for the NOESY experiments included in the supplementary material. | We thank the reviewer for revising the manuscript and for the kind suggestions and comments. Mixing times for the ROESY and NOESY experiments are 200 and 300 msec, respectively. But, we corrected L158 and L243 with a slightly different way. |
line 21 replace "compounds with "the compounds." line 38 replace "have been originally discovered" with "were originally isolated." line 85 replace "HMBC" with "the HMBC" line 99 replace "has been " with "was" line 110 and 126 replace "molecule" with "molecular" line 122 replace "NOE" with "an NOE" line 127 replace "for " with "of" line 158 replace "This" with "These" line 192 replace "frozen-dried" with "freeze-dried" line 192 replace ", filtered" with "and filtered" line 193 replace "over" with "by" line 219 replace "for the analysis" with "were used for the analysis" line 242 remove italic formatting on "inhabiting" line 243 replace "This" with "These" | All indicated corrections and suggested replacements have been followed. |

Reviewer 3 Report
This work is good-presented example of classical NPC study. In general, it may be recommended for publication in Marine Drugs, but after the revision. I'm not absolutely sure about suggested configurations of 6-OH in 2 and 5-OH in 3 and think that this stereocentres should be checked (verified) using strong proofs (e.g. Mosher's method). These evidences can highly improve the manuscript quality. As I understand, amounts of isolated compounds are enough for such study.
Several minor additional comments can be founded in the attached file.

Author Response
We thank the reviewer for raising the following points and for his valuable comments and kind suggestions.
Comments | Responses |
This work is good-presented example of classical NPC study. In general, it may be recommended for publication in Marine Drugs, but after the revision. I'm not absolutely sure about suggested configurations of 6-OH in 2 and 5-OH in 3 and think that this stereocentres should be checked (verified) using strong proofs (e.g. Mosher's method). These evidences can highly improve the manuscript quality. As I understand, amounts of isolated compounds are enough for such study | We have revised this part to further validate the configurations of 6-OH in 2 and 5-OH in 3 as illustrated in the manuscript. This was achieved and supported by comparison of the NMR chemical shifts of C-6, C-7, and H-6 of 2 with those of allied eunicellin-derived compounds (Figure 4) and/or by a molecular modeling study. We thought that all the information could establish the structure of 2. |

Round 2
Reviewer 1 Report
Dear Sirs,
Review concerns the Manuscript ID: marinedrugs-433354. Previously, due to low quality of Figures presented in Supplementary I was not able to review this manuscript. The corrected version contains some improvement. According to my remarks, the 13 NMR spectrum for the compound 2 was added. Data presented in Table 1 was corrected. However, Figures S2 and S9 (1H NMR for the compounds 1 and 2) were not improved. The chemical shifts of majority signals are not presented. Therefore, it is still difficult to check information based on which, the conclusion are drawn. The quality of Figures in Supplementary are still low. The improvement is really simple, so it is difficult to understand to me why it was not done.
However, despite of these difficulties, in my opinion, the structure determination is correct, therefore, the manuscript can be accepted.
Author Response
Table of responses to reviewers’ comments and suggestions
We thank the reviewer for his valuable comments and kind suggestions.
Responses to Reviewer 1
Comments | Responses |
Previously, due to low quality of Figures presented in Supplementary I was not able to review this manuscript. The corrected version contains some improvement. According to my remarks, the 13 NMR spectrum for the compound 2 was added. Data presented in Table 1 was corrected. However, Figures S2 and S9 (1H NMR for the compounds 1 and 2) were not improved. The chemical shifts of majority signals are not presented. Therefore, it is still difficult to check information based on which, the conclusion are drawn. The quality of Figures in Supplementary are still low. The improvement is really simple, so it is difficult to understand to me why it was not done.
However, despite of these difficulties, in my opinion, the structure determination is correct, therefore, the manuscript can be accepted | We have replaced the low quality 1H NMR spectra of compound 1 and 2 in Figures S2 and S10 by the high quality version. However, the detailed chemical shifts of each spectrum are illustrated in two sections in Figures S3 and S11, respectively. |

Reviewer 3 Report
The revised manuscript is satisfactory
Author Response
Table of responses to reviewers’ comments and suggestions
We thank the reviewer for his valuable comments and kind suggestions.
Comments | Responses |
The revised manuscript is satisfactory. | We thank the reviewer for his kind evaluation and judgment. |
